# COVID-19 disease—Temporal analyses of complete blood count parameters over course of illness, and relationship to patient demographics and management outcomes in survivors and non-survivors: A longitudinal descriptive cohort study

Simone Lanini[1]*, Chiara Montaldo[1], Emanuele Nicastri[1], Francesco Vairo[1], Chiara Agrati[1], Nicola Petrosillo[1], Paola Scognamiglio[1], Andrea Antinori[1], Vincenzo Puro[1], Antonino Di Caro[1], Gabriella De Carli[1], Assunta Navarra[1], Alessandro Agresta[1], Claudia Cimaglia[1], Fabrizio Palmieri[1], Gianpiero D'Offizi[1], Luisa Marchioni[1], Gary Pignac Kobinger[2], Markus Maeurer[3,4], Enrico Girardi[1], Maria Rosaria Capobianchi[1], Alimuddin Zumla[5,6], Franco Locatelli[7], Giuseppe Ippolito[1]

**1** National Institute for Infectious Diseases, Lazzaro Spallanzani, IRCCS, Rome, Italy, **2** Centre de Recherche en Infectiologie de l'Université Laval, Quebec City, QC, Canada, **3** Champalimaud Centre for the Unknown, Lisbon, Portugal, **4** University of Mainz, Mainz, Germany, **5** Department of Infection, Division of Infection and Immunity, University College London, London, United Kingdom, **6** NHS Foundation Trust, London, United Kingdom, **7** Department of Pediatric Hematology and Oncology IRCCS Ospedale Pediatrico Bambino Gesù, Rome, Italy

☯ These authors contributed equally to this work.

* simone.lanini@inmi.it

## Abstract

### Background

Detailed temporal analyses of complete (full) blood count (CBC) parameters, their evolution and relationship to patient age, gender, co-morbidities and management outcomes in survivors and non-survivors with COVID-19 disease, could identify prognostic clinical biomarkers.

### Methods

From 29 January 2020 until 28 March 2020, we performed a longitudinal cohort study of COVID-19 inpatients at the Italian National Institute for Infectious Diseases, Rome, Italy. 9 CBC parameters were studied as continuous variables [neutrophils, lymphocytes, monocytes, platelets, mean platelet volume, red blood cell count, haemoglobin concentration, mean red blood cell volume and red blood cell distribution width (RDW %)]. Model-based punctual estimates, as average of all patients' values, and differences between survivors and non-survivors, overall, and by co-morbidities, at specific times after symptoms, with relative 95% CI and P-values, were obtained by marginal prediction and ANOVA- style joint tests. All analyses were carried out by STATA 15 statistical package.

**Data Availability Statement:** All relevant data are within the paper and its Supporting Information files.

**Funding:** This work was supported by Line one-Ricerca Corrente 'Infezioni Emergenti e Riemergenti' and by Progetto COVID 2020 12371675 both funded by Italian Ministry of Health; AIRC (IG2018-21880); Regione Lazio (Gruppi di ricerca, E56C18000460002). The funders had no role in study design, data collection and analysis, decision to publish, or preparation of the manuscript.

**Competing interests:** The authors have declared that no competing interests exist.

## Main findings

379 COVID-19 patients [273 (72% were male; mean age was 61.67 (SD 15.60)] were enrolled and 1,805 measures per parameter were analysed. Neutrophils' counts were on average significantly higher in non-survivors than in survivors (P<0.001) and lymphocytes were on average higher in survivors (P<0.001). These differences were time dependent. Average platelets' counts (P<0.001) and median platelets' volume (P<0.001) were significantly different in survivors and non-survivors. The differences were time dependent and consistent with acute inflammation followed either by recovery or by death. Anaemia with anisocytosis was observed in the later phase of COVID-19 disease in non-survivors only. Mortality was significantly higher in patients with diabetes (OR = 3.28; 95%CI 1.51–7.13; p = 0.005), obesity (OR = 3.89; 95%CI 1.51–10.04; p = 0.010), chronic renal failure (OR = 9.23; 95%CI 3.49–24.36; p = 0.001), COPD (OR = 2.47; 95% IC 1.13–5.43; p = 0.033), cardiovascular diseases (OR = 4.46; 95%CI 2.25–8.86; p = 0.001), and those >60 years (OR = 4.21; 95%CI 1.82–9.77; p = 0.001). Age (OR = 2.59; 95%CI 1.04–6.45; p = 0.042), obesity (OR = 5.13; 95%CI 1.81–14.50; p = 0.002), renal chronic failure (OR = 5.20; 95%CI 1.80–14.97; p = 0.002) and cardiovascular diseases (OR 2.79; 95%CI 1.29–6.03; p = 0.009) were independently associated with poor clinical outcome at 30 days after symptoms' onset.

## Interpretation

Increased neutrophil counts, reduced lymphocyte counts, increased median platelet volume and anaemia with anisocytosis, are poor prognostic indicators for COVID19, after adjusting for the confounding effect of obesity, chronic renal failure, COPD, cardiovascular diseases and age >60 years.

## Background

Since the first report of SARS-CoV-2 as a novel human zoonotic pathogen in late December, 2019 [1], from Wuhan (China), the Coronavirus disease 2019 (COVID-19) pandemic [2] has rapidly spread worldwide. As of June 21st 2020, there have been 8,525,042 cases of COVID-19 with 456,973 deaths reported to the WHO [3]. Whilst COVID-19 predominantly affects the respiratory system, it is a multisystem disease, with a wide spectrum of clinical presentations from asymptomatic, mild and moderate, to severe, fulminant disease. The elderly and those with underlying co-morbidities such as cardiovascular disease, diabetes, chronic respiratory diseases and cancer appear to have increased risk of death [4]. As underlined by a recent review, COVID-19 has a significant impact on the hematopoietic system: lymphopenia, neutrophil/lymphocyte ratio and peak platelet/lymphocyte ratio may be considered as cardinal laboratory findings, with prognostic potential [5]. Complete blood count (CBC) is a routine investigation for all inpatients and provides vital parameters which can inform clinical management, including diagnosis, presence of infection or inflammation, anaemia, response to treatment, pathogenesis and stage of an inflammatory process. A study conducted in five different countries in Asia, Africa and United States shows that CBC is the most commonly used initial laboratory test [6]. A retrospective study from Singapore concluded that clinical findings and basic blood tests may be useful in identifying individuals with a higher probability of having COVID-19 [7].

To determine the temporal evolution of CBC parameters over the course of COVID-19 illness in survivors and non-survivors, and their potential association with patient clinical management outcomes, we performed an in depth analysis of routinely collected clinical and laboratory data of inpatients with COVID-19 disease admitted to the largest Italian COVID-19 clinical centre since the beginning of the Italian outbreak.

## Methods

### Study design

We conducted a longitudinal cohort study of all consecutive patients with laboratory confirmed COVID-19 admitted at the Italian National Institute for Infectious Diseases "Lazzaro Spallanzani" (INMI), in Rome (Italy) since the first Italian cases identified in Rome on January 29th, 2020.

### Setting

INMI is the largest Italian hospital for infectious diseases. Since the start of COVID-19 epidemic, INMI has been coordinating the clinical network for COVID-19 care in Lazio, an administrative Region of Italy with about 6 million inhabitants, whose main city is Rome. As regional referral centre, INMI is endowed with state-of-art laboratory and clinical facilities to care for COVID-19 patients at any stage of the disease. Thus, INMI may receive patients from other clinical centres but discharges them only after clinically recovery or death. Moreover, INMI runs the Regional Service for Surveillance of Infectious Diseases (SERESMI), which daily receives the notifications of all the new confirmed COVID-19 cases reported throughout the Region.

### Ethics/IRB approval

This study was approved by the IRB of INMI, in Rome (Italy).

### Patients and follow-up

All patients with confirmed COVID-19 admitted to INMI between the 29 January [8] and 28 March 2020 were followed up until death or discharge from hospital. Patients' demographics, clinical features and laboratory tests results were collected on standardised forms.

### Variables studied

We analysed nine CBC parameters (as continuous variables) including: neutrophils' count (cells/mm$^3$), lymphocytes' count (cells/mm$^3$), monocytes' count (cells/mm$^3$), platelets' count (cells/mm$^3$), mean platelet volume (MPV; fL), red blood cells' count (RBC; cells/mm$^3$), haemoglobin concentration (Hb; g/dL), mean red blood cell volume (MCV; fL) and red blood cell distribution width (RDW %). Values of all these parameters are recorded for 21 days since symptoms onset.

Other variables (covariates) included: patients life status at 30 days after symptoms' onset (binary, either survivors or non-survivors); time since symptoms' onset (as a continuous variable in day); age (binary; <60 or older); sex (binary); obesity (binary, body max index >30 or lower); chronic renal failure; cardiovascular disease (including hypertension); diabetes; history of cancer throughout life and chronic obstructive pulmonary disease (COPD).

## Definitions

Survivors were defined as patients who recovered and were discharged from hospital or who were still hospitalized within 30 days after symptoms onset.

Non-survivors are all those subjects who died within 30 days after onset of symptoms.

## Data collection and data quality assessment

All patients' demographic, laboratory and clinical data (CBC parameter values, day of hospital admission, day of patients' discharge and patient's life status at discharge) were collected in the Regional Service for the Epidemiology and surveillance of Infectious Diseases database.

## Laboratory methods

SARS-CoV-2 RNA was extracted by QIAsymphony (QIAgen, Germany) and Real-time reverse-transcription PCR (RT-PCR) targeting E and RdRp viral genes was used to assess the presence of SARS-CoV-2 RNA, as per Corman protocol [9].

CBC was performed using the fully automated haematology analyser Sismex XN 1500.

## Statistics and modelling

All data analyses were carried out by STATA 15 statistical package (S1 File).

The temporal kinetics of the nine blood counts parameters were modelled according to separate mixed effect multivariable linear-log regression models (MEMLR) [10,11]. Overall, we modelled a total of 1,805 measures per parameter taken from the 379 subjects in follow-up. The average number of measures per patients was 4.8 (range 1–22) and the average number of observation per day was 82.0 (range 9–122)). For each one of the nine analysed CBC parameters, we provided a plot for describing parameter variation over time (kinetic), either for survivors or for non-survivors. The random component of each model included a random intercept at patient's level, a random slope at the time since symptoms' onset and an unstructured covariance matrix for dealing with measures imbalance over time (i.e. unequal number of observation per patient). The fixed component included one dependent variable (i.e. one of nine blood counts parameters in natural log-form), two main independent variables for modelling kinetics (i.e. patient's clinical outcome and time since symptoms' onset) and a set of potential confounders (i.e. the set of variables with a significant association with clinical outcome in previous MVLR model).

The kinetics of each CBC parameter was modelled assuming time since symptoms' onset as either linear or quadratic component. Thus, we explored four different hypothetical shapes of associations (i.e. liner positive, linear negative, U-shaped and inverse U-shaped trajectory). Linear association was preferred over quadratic association whenever LRT was <0.100 (assuming linear MEMLR nested into quadratic MEMLR). A full interaction term between time since symptoms' onset and clinical outcome was used so that the kinetics of each CBC parameter could have a different temporal trend either in survivors or in non-survivors.

We obtained model-based punctual estimates of CBC parameters at different times and difference between survivors and non-survivors as overall and at specific time after symptoms' onset with relative 95% CI and P-values, by marginal prediction and ANOVA- style joint tests. Estimate values of CBC parameters and relative difference were always shown in exponential form for simplicity. Multivariable logistic regression (MVLR) models were carried out for assessing the association between clinical outcome and patients' demographics. All kinetic models were adjusted for the four potential confounders selected in the MVLR (i.e. age, obesity, chronic renal failure and cardiovascular disease).

## Results

### Patients recruited and demographics

Fig 1 shows study patients' flowchart. Four-hundred-ten patients were admitted with a diagnosis of COVID-19 at INMI Spallanzani between January 29 and March 28. Of them, 379 patients were included in the analyses. Thirty patients (7.32%) were excluded due to incomplete records and one died before CBC was performed. Forty-one patients died within 30 days after symptoms' onset with a case fatality rate of 10.82%. Among the 379 patients included in the analysis, 273 (72.03%) were male and the mean age was 61.67 (SD 15.60).

Patients' demographics and co-morbidities are shown in Table 1.

### Association between patients baseline conditions and clinical outcome

Bivariable analyses showed that mortality was significantly higher in patients aged 60 years or older (OR = 4.21; 95%CI 1.82–9.77; p = 0.001), in diabetic patients (OR = 3.28; 95%CI 1.51–7.13; p = 0.005), in obese patients (OR = 3.89; 95%CI 1.51–10.04; p = 0.010), in patients with underlying chronic renal failure (OR = 9.23; 95%CI 3.49–24.36; p = 0.001), in patients with COPD (OR = 2.47; 95% IC 1.13–5.43; p = 0.033) and in patients with cardiovascular diseases (OR = 4.46; 95%CI 2.25–8.86; p = 0.001). No association was found between clinical outcome and sex, and between clinical outcome and history of neoplasm (defined as any anamnestic data of oncological disease).

The multivariable analysis showed that only age (OR = 2.59; 95%CI 1.04–6.45; p = 0.042), obesity (OR = 5.13; 95%CI 1.81–14.50; p = 0.002), renal chronic failure (OR = 5.20; 95%CI 1.80–14.97; p = 0.002) and cardiovascular diseases (OR 2.79; 95%CI 1.29–6.03; p = 0.009) were independently associated with patient clinical outcome at 30 day after symptoms onset. Table 1 reports the results of bivariable and multivariable analysis. Additional S2 File reports the details of the model building and variable selection process.

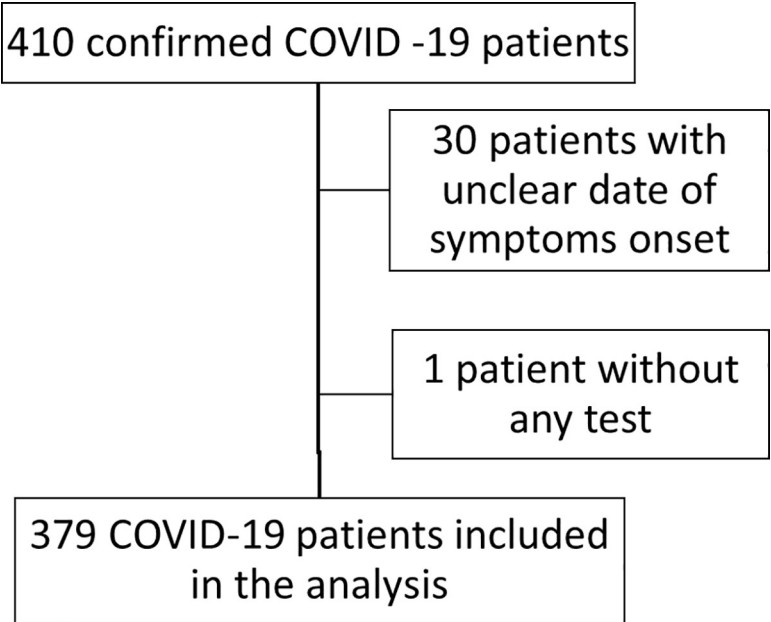

**Fig 1. Selection of patients.** Between January 29, 2020 and March 28, 2020, 410 patients tested positive to SARSCoV-2 RT-PCR were admitted in Spallanzani Hospital, of whom 379 were included in the analysis. For the 379 patients included in the analysis a total of 1,805 measurable haematological test results were available.

Table 1. Patients' demographics and co-morbidities.

| Patients feature | | Descriptive analysis | | | | | | Bivaribale analysis | | | | Multivariable analysis | | | |
|---|---|---|---|---|---|---|---|---|---|---|---|---|---|---|---|
| | | Total | | survivors | | Non-survivors | | OR | 95% CI | | p-value | OR | 95% CI | | p-value |
| | | N | % | N | % | N | % | | | | | | | | |
| Age | <60 years | 164 | 43.27% | 157 | 95.73% | 7 | 4.27% | 4.21 | 1.82 | 9.77 | 0.001 | 2.59 | 1.04 | 6.45 | 0.042 |
| | >60 years | 215 | 56.73% | 181 | 84.19% | 34 | 15.81% | | | | | | | | |
| Sex | female | 106 | 27.97% | 93 | 87.74% | 13 | 12.26% | 0.82 | 0.41 | 1.65 | 0.577 | - | - | - | - |
| | male | 273 | 72.03% | 245 | 89.74% | 28 | 10.26% | | | | | | | | |
| Diabetes | no | 334 | 88.13% | 304 | 91.02% | 30 | 8.98% | 3.28 | 1.51 | 7.13 | 0.005 | - | - | - | - |
| | yes | 45 | 11.87% | 34 | 75.56% | 11 | 24.44% | | | | | | | | |
| Neoplasm | no | 360 | 94.99% | 322 | 89.44% | 38 | 10.56% | 1.59 | 0.44 | 5.70 | 0.498 | - | - | - | - |
| | yes | 19 | 5.01% | 16 | 84.21% | 3 | 15.79% | | | | | | | | |
| Obesity | no | 355 | 93.67% | 321 | 90.42% | 34 | 9.58% | 3.89 | 1.51 | 10.04 | 0.010 | 5.13 | 1.81 | 14.50 | 0.002 |
| | yes | 24 | 6.33% | 17 | 70.83% | 7 | 29.17% | | | | | | | | |
| Chronic renal failure | no | 360 | 94.99% | 328 | 91.11% | 32 | 8.89% | 9.23 | 3.49 | 24.36 | 0.001 | 5.20 | 1.80 | 14.97 | 0.002 |
| | yes | 19 | 5.01% | 10 | 52.63% | 9 | 47.37% | | | | | | | | |
| COPD | no | 330 | 87.07% | 299 | 90.61% | 31 | 9.39% | 2.47 | 1.13 | 5.43 | 0.033 | - | - | - | - |
| | yes | 49 | 12.93% | 39 | 79.59% | 10 | 20.41% | | | | | | | | |
| Cardiovascular disease | no | 250 | 65.96% | 236 | 94.40% | 4 | 1.60% | 4.46 | 2.25 | 8.86 | 0.001 | 2.79 | 1.29 | 6.03 | 0.009 |
| | yes | 129 | 34.04% | 102 | 79.07% | 27 | 20.93% | | | | | | | | |
| Overall | - | 379 | 100.00% | 338 | 89.18% | 41 | 10.82% | - | - | - | - | - | - | - | - |

## Leukocytes parameters kinetics

Three leukocyte parameters, neutrophils' counts, lymphocytes' counts and monocytes' counts, were analysed (**Fig 2** and **Table 2**). Average neutrophils' counts (**Fig 2A and 2B**) were significantly higher in non-survivors than in survivors (p<0.001). The temporal analysis suggested that, at the time of symptoms' onset, survivors and non-survivors had similar level of neutrophil counts (p = 0.191). In survivors, neutrophils' counts remained steadily and within the normal range throughout the follow-up. In contrast, in non-survivors, neutrophils' counts sharply increased, being significantly different from those of survivors by day 6 after symptoms' onset. Moreover, the model predicted that by day 13, the average neutrophils' counts in non-survivors steadily exceeded the upper normal limit range (8,000 cells/mm$^3$). Among the considered confounders, only age was significantly associated to higher neutrophil counts (p = 0.027), while no significant association was found for obesity, chronic renal failure and cardiovascular diseases.

Average lymphocytes' counts (**Fig 2C and 2D**) were significantly lower in non-survivors than in survivors (p-joint <0.001) since the first day of symptoms' onset. The model predicted that lymphocytes' counts were significantly lower in non-survivors than in survivors since the first day after symptoms' onset (p = 0.033). In particular, the difference of lymphocytes' counts between survivors and non-survivors widened over time, being estimated at 330.1 (95% CI 633.48–26.85; p = 0.033) and 765.73 (95% CI 1050.04–481.43; p<0.001) cells per mm$^3$ at symptoms' onset and at the end of follow-up, respectively. Moreover, average lymphocytes' count was always below 1,000 cells per mm$^3$ in non-survivors. In contrast, the level of lymphocytes normalized by the day 15 in survivors. Among the considered confounders, age and chronic renal failure were significantly associated to lower lymphocyte counts (respectively p<0.001, and p = 0.052) while no significant association was found for obesity and cardiovascular diseases.

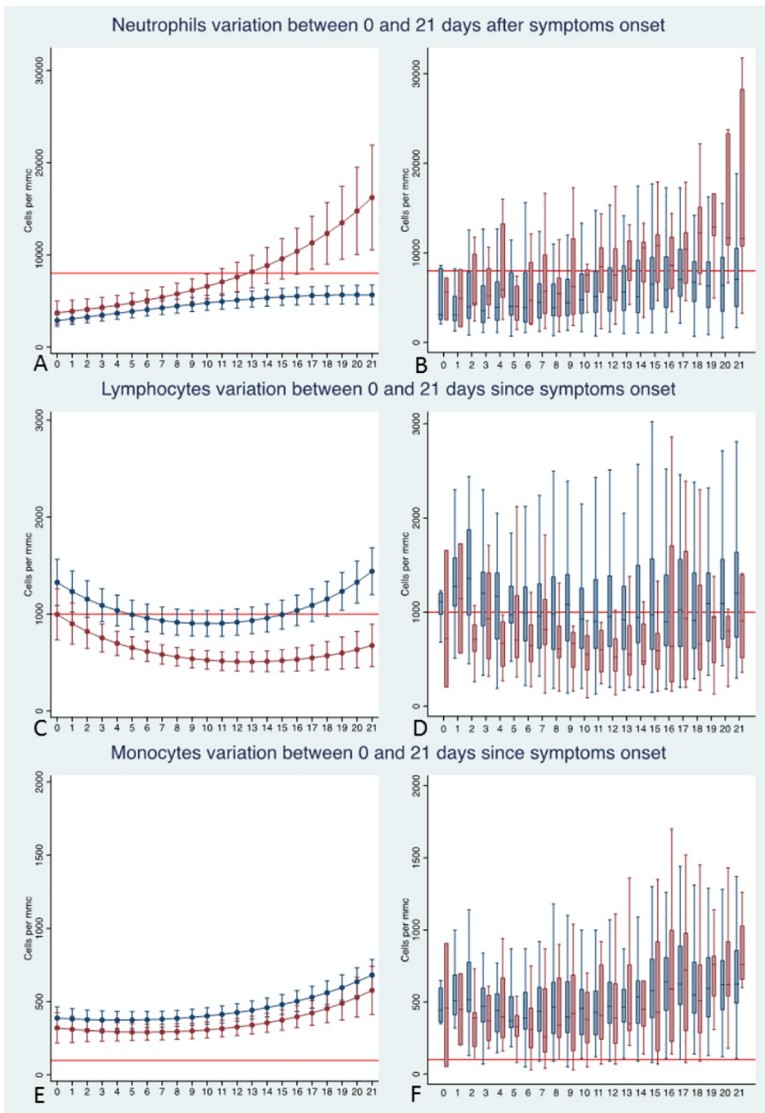

**Fig 2.** Plots of predicted values and box plots of observed values of average neutrophils (A, B), lymphocytes (C, D) and monocytes (E, F) counts over time for survivors (blue line and boxes) and non-survivors (red line and boxes). In the box plots the outliers are not displayed. All estimates were made on the full dataset, including 1805 determinations on the 379 patients. The red line in plots A and B marks the upper normal limit range of neutrophils (8,000 cells/mmc (red line). The red line in plots C and D marks the lower normal limit of lymphocytes (1,000 cells/mmc). The red line in plot E and F marks the lower upper normal limit of monocytes (100 cells/mmc)

Average monocytes' counts (**Fig 2E and 2F**) were marginally lower in non-survivors than in survivors (p-joint <0.058). The temporal analysis suggested that average variation of monocytes was within the normal range in both survivors and non-survivors. Among the considered confounders, only age was significantly associated to lower monocytes' counts (p = 0.038) while no significant association was found for obesity, chronic renal failure and cardiovascular diseases.

## Red blood cells parameters and kinetics

Four RBC parameters (i.e. RBC count, Hb concentration, MCV, RDW) were analysed (**Fig 3 and Table 3**). Average RBC (**Fig 3A and 3B**) were significantly lower in non-survivors than in

**Table 2. Temporal evolution of the differences of neutrophils', lymphocyte's and monocytes' counts, between survivors and non-survivors (diff).**

| Day | Neutrophils' counts (cells/mm3) | | | | Lymphocytes' counts (cells/mm3) | | | | Monocytes' counts (cells/mm3) | | | |
|---|---|---|---|---|---|---|---|---|---|---|---|---|
| | Diff | llb | ulb | p | Diff | llb | ulb | p | diff | llb | ulb | p |
| 0 | 841.22 | -419.16 | 2101.60 | 0.191 | -330.16 | -633.48 | -26.85 | 0.033 | -66.94 | -183.12 | 49.25 | 0.259 |
| 1 | 821.01 | -351.81 | 1993.82 | 0.170 | -332.25 | -578.06 | -86.44 | 0.008 | -70.75 | -170.66 | 29.15 | 0.165 |
| 2 | 816.62 | -281.34 | 1914.58 | 0.145 | -333.86 | -536.93 | -130.80 | 0.001 | -74.18 | -160.95 | 12.60 | 0.094 |
| 3 | 831.14 | -207.17 | 1869.46 | 0.117 | -335.53 | -507.59 | -163.47 | <0.001 | -77.30 | -153.77 | -0.82 | 0.048 |
| 4 | 868.00 | -128.49 | 1864.49 | 0.088 | -337.66 | -488.02 | -187.30 | <0.001 | -80.18 | -148.90 | -11.47 | 0.022 |
| 5 | 930.94 | -43.81 | 1905.69 | 0.061 | -340.60 | -476.49 | -204.70 | <0.001 | -82.89 | -146.11 | -19.67 | 0.010 |
| 6 | 1024.11 | 49.55 | 1998.67 | 0.039 | -344.61 | -471.51 | -217.72 | <0.001 | -85.46 | -145.14 | -25.78 | 0.005 |
| 7 | 1152.04 | 155.75 | 2148.32 | 0.023 | -349.96 | -471.86 | -228.07 | <0.001 | -87.93 | -145.69 | -30.18 | 0.003 |
| 8 | 1319.68 | 280.33 | 2359.03 | 0.013 | -356.89 | -476.64 | -237.15 | <0.001 | -90.33 | -147.44 | -33.22 | 0.002 |
| 9 | 1532.47 | 429.70 | 2635.23 | 0.007 | -365.64 | -485.28 | -245.99 | <0.001 | -92.67 | -150.10 | -35.24 | 0.002 |
| 10 | 1796.35 | 610.43 | 2982.26 | 0.003 | -376.46 | -497.56 | -255.37 | <0.001 | -94.96 | -153.44 | -36.48 | 0.002 |
| 11 | 2117.84 | 828.49 | 3407.20 | 0.001 | -389.65 | -513.48 | -265.83 | <0.001 | -97.20 | -157.30 | -37.09 | 0.002 |
| 12 | 2504.12 | 1088.64 | 3919.61 | 0.001 | -405.53 | -533.30 | -277.76 | <0.001 | -99.37 | -161.61 | -37.12 | 0.002 |
| 13 | 2963.08 | 1393.91 | 4532.26 | <0.001 | -424.48 | -557.51 | -291.45 | <0.001 | -101.44 | -166.42 | -36.46 | 0.002 |
| 14 | 3503.46 | 1745.19 | 5261.74 | <0.001 | -446.96 | -586.79 | -307.13 | <0.001 | -103.38 | -171.89 | -34.86 | 0.003 |
| 15 | 4134.97 | 2140.90 | 6129.04 | <0.001 | -473.49 | -622.04 | -324.94 | <0.001 | -105.12 | -178.34 | -31.90 | 0.005 |
| 16 | 4868.41 | 2576.86 | 7159.96 | <0.001 | -504.73 | -664.45 | -345.02 | <0.001 | -106.59 | -186.26 | -26.92 | 0.009 |
| 17 | 5715.91 | 3046.30 | 8385.52 | <0.001 | -541.46 | -715.48 | -367.44 | <0.001 | -107.67 | -196.30 | -19.04 | 0.017 |
| 18 | 6691.13 | 3539.92 | 9842.34 | <0.001 | -584.63 | -776.99 | -392.28 | <0.001 | -108.22 | -209.31 | -7.13 | 0.036 |
| 19 | 7809.51 | 4045.91 | 11573.11 | <0.001 | -635.40 | -851.24 | -419.56 | <0.001 | -108.04 | -226.24 | 10.16 | 0.073 |
| 20 | 9088.61 | 4549.64 | 13627.59 | <0.001 | -695.19 | -941.08 | -449.29 | <0.001 | -106.90 | -248.20 | 34.40 | 0.138 |
| 21 | 10548.49 | 5033.08 | 16063.90 | <0.001 | -765.73 | -1050.04 | -481.43 | <0.001 | -104.47 | -276.42 | 67.49 | 0.234 |
| Joint | | | | <0.001 | | | | <0.001 | | | | 0.058 |

llb = low limit bound, ulb = lower limit bound.

survivors (p-joint <0.001). The temporal analysis suggested that, at the time of symptoms' onset, survivors and non-survivors had similar RBC counts (p = 0.257). The model predicted that average level of RBC became significantly different between survivors and non-survivors by day 2 after symptoms' onset (p = 0.033). Average RBC in non-survivors steadily decreased over time being below to normal value (3.8 million cell/mm$^3$) by day 7 after symptoms onset. In contrast, we found no evidence for a significant decrease of average RBC counts below normal level in survivors. Among the considered confounders, age (p<0.001) and chronic renal failure (p<0.001) were significantly associated to lower RBC counts. No significant association was found for obesity and cardiovascular diseases.

Average Hb levels (**Fig 3A and 3B**) were significantly lower in non-survivors than in survivors (p-joint <0.001). The temporal analysis suggested that, at the time of symptoms' onset, the difference between survivors and non-survivors was marginal (p = 0.063) but became statistically significant by the next day (p = 0.041). The average levels of Hb were steadily within the normal range in survivors while a mild (non-statistically significant) anaemia (Hb level <10 mg/dL) was observed at the end of follow-up in non-survivors. Among the considered confounders, age (p<0.001) and chronic renal failure (p<0.001) were significantly associated to lower Hb level. No significant association was found for obesity and cardiovascular diseases.

Average MCVs (**Fig 3C and 3D**) were significantly higher in non-survivors than in survivors (p-joint <0.001). The temporal analysis suggested that, at the time of symptoms' onset, MCV difference between survivors and non-survivors was marginal (p = 0.066) but became

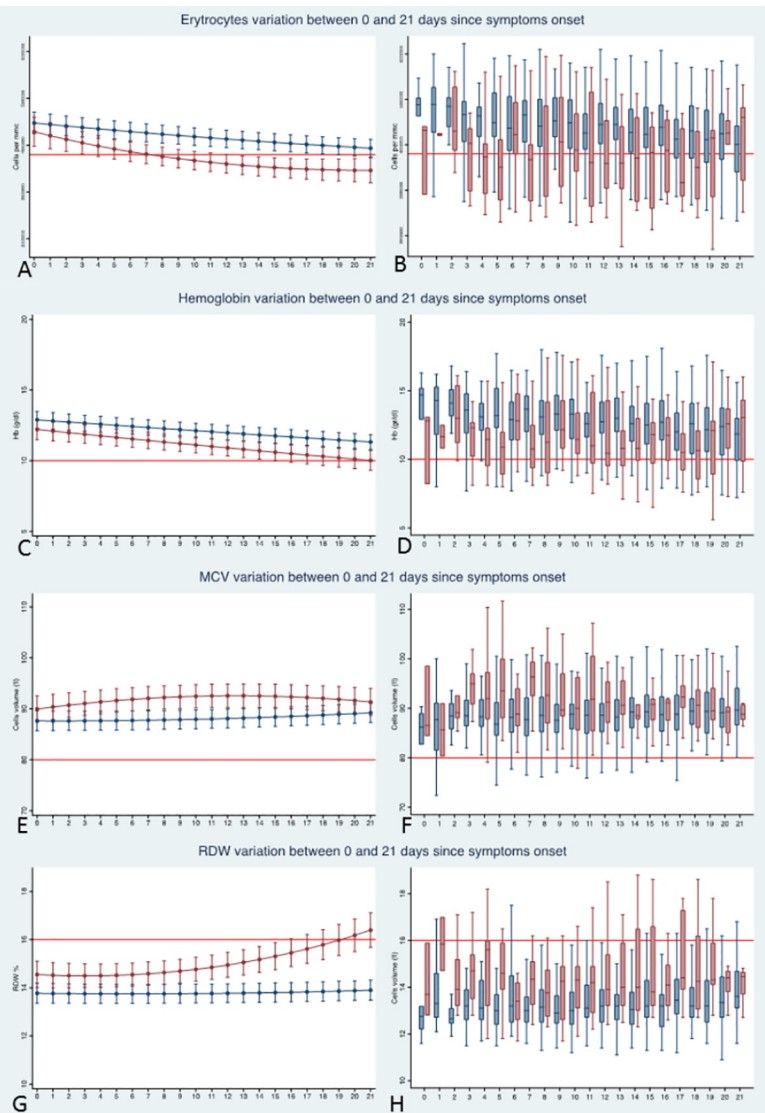

**Fig 3.** Plots of predicted values and box plots of observed values of average erythrocytes counts (A, B), Hb level (C, D), MCV (E, F) and RDW percentage (D, H) over time for survivors (blue line and boxes) and non-survivors (red line and boxes). In the box plots, the outliers are not displayed. All estimates were made on the full dataset, including 1805 determinations on the 379 patients. The red line in plots A and B marks the lower normal limit of erythrocytes (3.8 million cell/mmc). The red line in plots C and D marks the lower normal limit of Hb (10 mg/dL). The red line in plots E and F marks the normal MCV value (80 fL). The red line in plot G and H marks the lower normal average percentages of RDW (16%).

statistically significant by the next day (p = 0.024). Although average of MCVs was higher in non-survivors than in survivors, the variations of MCV were always within the normal range in both survivors and non-survivors. Among the considered confounders, chronic renal failure was associated with higher MCV (p = 0.026) and cardiovascular diseases were associated with lower MCV (p = 0.013), while no significant association was found for age and obesity.

Average RDW percentages (**Fig 3E and 3F**) were significantly higher in non-survivors than in survivors (p-joint <0.001). The temporal analysis suggested that this difference was already significant since the first day of symptoms' onset (p = 0.004). The average percentages of RDW were steadily within the normal range in survivors while a mild (non-statistically significant)

**Table 3. Temporal evolution of the difference of red cell count, Haemoglobin level, MCV and RDW between survivors and non-survivors (diff).**

| | RBC counts (cells/mm3) | | | | Hb (g/dL) | | | | MCV (fL) | | | | RDW (%) | | | |
|---|---|---|---|---|---|---|---|---|---|---|---|---|---|---|---|---|
| Day | diff | llb | ulb | p | diff | llb | ulb | p | diff | llb | ulb | p | diff | llb | ulb | p |
| 0 | -186987.30 | -510433.40 | 136458.70 | 0.257 | -0.68 | -1.40 | 0.04 | 0.063 | 2.34 | -0.15 | 4.83 | 0.066 | 0.78 | 0.25 | 1.31 | 0.004 |
| 1 | -238952.20 | -529112.80 | 51208.26 | 0.107 | -0.72 | -1.41 | -0.03 | 0.041 | 2.74 | 0.36 | 5.12 | 0.024 | 0.76 | 0.26 | 1.26 | 0.003 |
| 2 | -286012.50 | -548989.80 | -23035.31 | 0.033 | -0.75 | -1.41 | -0.09 | 0.025 | 3.10 | 0.81 | 5.39 | 0.008 | 0.74 | 0.27 | 1.22 | 0.002 |
| 3 | -328412.70 | -569877.30 | -86948.04 | 0.008 | -0.79 | -1.42 | -0.16 | 0.015 | 3.42 | 1.20 | 5.64 | 0.003 | 0.74 | 0.28 | 1.20 | 0.002 |
| 4 | -366377.20 | -591452.70 | -141301.60 | 0.001 | -0.82 | -1.43 | -0.21 | 0.008 | 3.70 | 1.53 | 5.87 | 0.001 | 0.75 | 0.30 | 1.20 | 0.001 |
| 5 | -400112.00 | -613254.00 | -186970.10 | <0.001 | -0.85 | -1.44 | -0.26 | 0.005 | 3.94 | 1.81 | 6.07 | <0.001 | 0.77 | 0.33 | 1.21 | 0.001 |
| 6 | -429805.60 | -634713.90 | -224897.40 | <0.001 | -0.89 | -1.46 | -0.31 | 0.003 | 4.14 | 2.03 | 6.24 | <0.001 | 0.80 | 0.36 | 1.24 | <0.001 |
| 7 | -455630.00 | -655226.80 | -256033.10 | <0.001 | -0.92 | -1.48 | -0.36 | 0.001 | 4.30 | 2.20 | 6.39 | <0.001 | 0.84 | 0.40 | 1.27 | <0.001 |
| 8 | -477741.50 | -674224.50 | -281258.40 | <0.001 | -0.95 | -1.50 | -0.40 | 0.001 | 4.41 | 2.32 | 6.50 | <0.001 | 0.88 | 0.44 | 1.33 | <0.001 |
| 9 | -496282.10 | -691238.20 | -301325.90 | <0.001 | -0.98 | -1.53 | -0.43 | 0.001 | 4.49 | 2.40 | 6.58 | <0.001 | 0.94 | 0.49 | 1.39 | <0.001 |
| 10 | -511379.70 | -705935.60 | -316823.90 | <0.001 | -1.01 | -1.56 | -0.46 | <0.001 | 4.52 | 2.43 | 6.61 | <0.001 | 1.01 | 0.55 | 1.46 | <0.001 |
| 11 | -523149.30 | -718133.90 | -328164.60 | <0.001 | -1.04 | -1.59 | -0.49 | <0.001 | 4.51 | 2.41 | 6.61 | <0.001 | 1.09 | 0.62 | 1.55 | <0.001 |
| 12 | -531692.90 | -727797.30 | -335588.50 | <0.001 | -1.07 | -1.62 | -0.51 | <0.001 | 4.46 | 2.35 | 6.57 | <0.001 | 1.17 | 0.70 | 1.65 | <0.001 |
| 13 | -537101.00 | -735025.40 | -339176.60 | <0.001 | -1.10 | -1.66 | -0.53 | <0.001 | 4.37 | 2.24 | 6.49 | <0.001 | 1.27 | 0.79 | 1.76 | <0.001 |
| 14 | -539452.30 | -740038.00 | -338866.60 | <0.001 | -1.12 | -1.70 | -0.55 | <0.001 | 4.23 | 2.09 | 6.38 | <0.001 | 1.38 | 0.89 | 1.88 | <0.001 |
| 15 | -538814.60 | -743156.40 | -334472.90 | <0.001 | -1.15 | -1.74 | -0.56 | <0.001 | 4.06 | 1.89 | 6.22 | <0.001 | 1.51 | 0.99 | 2.02 | <0.001 |
| 16 | -535245.00 | -744780.50 | -325709.50 | <0.001 | -1.18 | -1.78 | -0.57 | <0.001 | 3.84 | 1.64 | 6.04 | 0.001 | 1.64 | 1.11 | 2.17 | <0.001 |
| 17 | -528790.10 | -745360.30 | -312219.80 | <0.001 | -1.21 | -1.83 | -0.58 | <0.001 | 3.58 | 1.34 | 5.81 | 0.002 | 1.78 | 1.23 | 2.34 | <0.001 |
| 18 | -519486.50 | -745361.60 | -293611.40 | <0.001 | -1.23 | -1.87 | -0.59 | <0.001 | 3.28 | 0.99 | 5.56 | 0.005 | 1.94 | 1.36 | 2.52 | <0.001 |
| 19 | -507360.90 | -745229.90 | -269491.80 | <0.001 | -1.26 | -1.92 | -0.60 | <0.001 | 2.93 | 0.58 | 5.28 | 0.014 | 2.11 | 1.51 | 2.72 | <0.001 |
| 20 | -492430.20 | -745358.90 | -239501.50 | <0.001 | -1.28 | -1.97 | -0.60 | <0.001 | 2.55 | 0.12 | 4.97 | 0.040 | 2.30 | 1.65 | 2.94 | <0.001 |
| 21 | -474701.90 | -746069.30 | -203334.50 | 0.001 | -1.31 | -2.01 | -0.60 | <0.001 | 2.12 | -0.41 | 4.64 | 0.100 | 2.49 | 1.81 | 3.18 | <0.001 |
| Joint | | | | <0.001 | | | | <0.001 | | | | <0.001 | | | | <0.001 |

llb = low limit bound, ulb = lower limit bound.

increase of anisocytosis (RDW >16%) was observed at the end of follow-up in non-survivors. Among the considered confounders, age, chronic renal failure and cardiovascular diseases were significantly associated to higher RDW percentage (p<0.001 for the three confounders) while no significant association was found for obesity.

## Platelets parameters and kinetics

Two platelets parameters (i.e. platelets' counts and MPV) were analysed (**Fig 4** and **Table 4**). Average platelets' counts (**Fig 4A and 4B**) were significantly different between non-survivors and survivors (p-joint <0.001). The temporal analysis suggested that platelets' counts were lower in survivors than in non-survivors at beginning of the diseases, while the opposite was evident at the end of the follow-up. In survivors, platelets' counts peaked at 302,758 cells/mm$^3$ (95% CI: 264,219–341,298) on day 19 and eventually remained steady. In contrast, platelets' counts in non-survivors peaked at 223,850 cells/mm$^3$ (95% CI: 190,457–257,242) on day 9 and eventually decreased, being significantly below the lower normal limit value (150,000 cells/mm$^3$) at the end of follow-up. Among confounders, only age was significantly associated to lower platelets' count (p = 0.001) while no significant association was found for obesity, chronic renal failure and cardiovascular diseases.

Average MPV (**Fig 4C and 4D**) had patterns of temporal variations similar to those observed for platelets' counts. MPV was significantly higher in survivors than in non-survivors

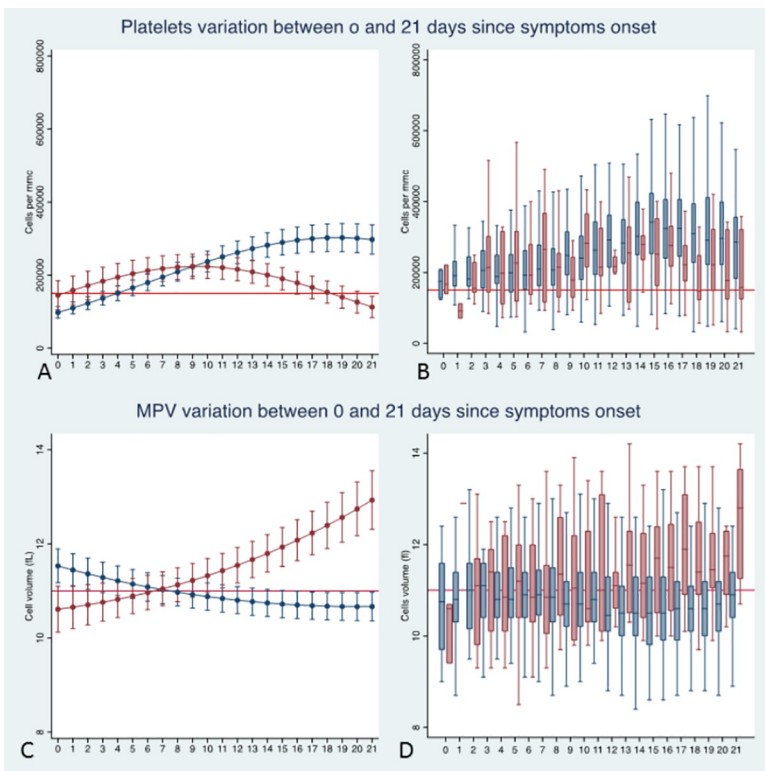

**Fig 4.** Plots of predicted values and box plots of observed values of average platelets counts (A, B) and MPV (C, D) over time for survivors (blue line and boxes) and non-survivors (red line and boxes). In the box plots the outliers are not displayed. All estimates were made on the full dataset, including 1805 determinations on the 379 patients. The red line in plots A and B marks the normal limit value of PLT (150,000 cells/mmc); the red line in plotd C and D marks the upper normal limit value of MPV (11 fl).

at the beginning of the disease (p<0.001) while the opposite was observed at the end of the follow-up, with MPV significantly higher in non–survivors than in survivors (p<0.001). MPV tended to normalize over time in survivors, it steadily increased in non-survivors, exceeding the upper normal limit value (i.e. 11 fl) by the day 7 after symptoms' onset. Among the considered confounders, only chronic renal failure is significantly associated to higher MPV (p = 0.022) while no significant association was found for age, obesity, and cardiovascular diseases.

## Discussion

We found that increased neutrophil counts, reduced lymphocyte counts, increased MPV, anaemia with anisocytosis, in association with obesity, chronic renal failure, COPD, cardiovascular diseases and age >60 years are poor prognostic indicators for COVID19.

*The first analysed CBC set* was leukocytes. Whilst in non-survivors lymphocytes' counts were persistently below the lower normal limit, in survivors lymphopenia was (on average) only mild and transient. This observation suggests that lymphopenia can be considered a surrogate biomarker of ineffective immune response to the SARS-CoV-2 [12] infection, as observed for other coronavirus including MERS [13] and SARS-CoV-1 [14]. The biological reasons associated to lymphopenia are under investigation and whilst this could be due to localisation at sites of disease migrating from peripheral blood, it may be also be due to dysregulation of the cytokine network [15]. Neutrophils' counts showed an opposite trend, significantly

**Table 4. Temporal evolution of the differences of platelets' counts and MPV, between survivors and non-survivors (diff).**

| Day | Platelets' counts (cells/mm3) | | | | MPV (fL) | | | |
|---|---|---|---|---|---|---|---|---|
| | diff | llb | ulb | p | diff | llb | ulb | p |
| 0 | 47337.03 | 7861.02 | 86813.05 | 0.019 | -0.92 | -1.43 | -0.40 | 0.001 |
| 1 | 48510.89 | 9369.23 | 87652.56 | 0.015 | -0.79 | -1.26 | -0.32 | 0.001 |
| 2 | 48464.84 | 9983.32 | 86946.37 | 0.014 | -0.66 | -1.10 | -0.22 | 0.003 |
| 3 | 46982.17 | 9414.44 | 84549.90 | 0.014 | -0.53 | -0.93 | -0.12 | 0.012 |
| 4 | 43871.02 | 7384.70 | 80357.34 | 0.018 | -0.39 | -0.78 | -0.01 | 0.045 |
| 5 | 38976.60 | 3645.00 | 74308.21 | 0.031 | -0.26 | -0.62 | 0.11 | 0.167 |
| 6 | 32192.67 | -2007.41 | 66392.76 | 0.065 | -0.12 | -0.47 | 0.23 | 0.500 |
| 7 | 23471.41 | -9712.96 | 56655.77 | 0.166 | 0.02 | -0.32 | 0.36 | 0.922 |
| 8 | 12831.03 | -19536.53 | 45198.58 | 0.437 | 0.16 | -0.18 | 0.49 | 0.360 |
| 9 | 360.46 | -31457.88 | 32178.80 | 0.982 | 0.30 | -0.03 | 0.63 | 0.079 |
| 10 | -13779.49 | -45366.39 | 17807.41 | 0.393 | 0.44 | 0.11 | 0.78 | 0.009 |
| 11 | -29358.67 | -61060.23 | 2342.90 | 0.070 | 0.59 | 0.25 | 0.93 | 0.001 |
| 12 | -46083.82 | -78250.46 | -13917.18 | 0.005 | 0.74 | 0.40 | 1.09 | <0.001 |
| 13 | -63608.35 | -96570.40 | -30646.29 | <0.001 | 0.89 | 0.54 | 1.25 | <0.001 |
| 14 | -81545.08 | -115590.40 | -47499.80 | <0.001 | 1.05 | 0.69 | 1.42 | <0.001 |
| 15 | -99481.31 | -134837.10 | -64125.49 | <0.001 | 1.21 | 0.83 | 1.59 | <0.001 |
| 16 | -116995.30 | -153816.80 | -80173.76 | <0.001 | 1.37 | 0.97 | 1.78 | <0.001 |
| 17 | -133673.30 | -172039.10 | -95307.43 | <0.001 | 1.54 | 1.11 | 1.97 | <0.001 |
| 18 | -149125.80 | -189040.40 | -109211.20 | <0.001 | 1.72 | 1.25 | 2.18 | <0.001 |
| 19 | -163003.00 | -204405.00 | -121601.00 | <0.001 | 1.89 | 1.39 | 2.39 | <0.001 |
| 20 | -175007.30 | -217781.90 | -132232.70 | <0.001 | 2.07 | 1.53 | 2.62 | <0.001 |
| 21 | -184903.40 | -228896.60 | -140910.30 | <0.001 | 2.26 | 1.66 | 2.86 | <0.001 |
| Joint | | | | <0.001 | | | | <0.001 |

llb = low limit bound, ulb = lower limit boun.

increasing during the second week of the disease in non-survivors, while they remained within the normal range in survivors. The strong association between patients' clinical outcome and increase of neutrophil counts was a biologically and clinically relevant finding of our study and the reasons for that are manifold. Although bacterial infections could not be ruled out in this study, the inverse relationship between neutrophils' and lymphocytes' counts with disease progression could have other explanations. *First*, the opposite kinetics in neutrophils versus lymphocytes dynamics might be driven by depletion of CD8+ circulating lymphocytes and a concomitant increased release of serum IL-6, IL-10, IL-2 and IFN-γ in patients with severe disease presentations as compared to patients with moderate disease [16]. *Second*, the high neutrophils may be due to a massive expansion of myeloid cells with suppressor activity [granulocytes myeloid suppressor cells-MDSC]. These cells are morphologically identical to neutrophils in standard laboratory examination but they are functionally distinct. MDSC are characterized by their myeloid origin, immature state, and by their potent ability to suppress T-cell functions and to modulate cytokines' production [17]. According to this hypothesis, the hyper inflammatory state observed in severe COVID-19 cases, may induce an abnormal expansion of MDSC that in turn strongly affect the immunological response required for viral clearance [18,19].

*The second CBC* set included platelets' counts and MPV. The variation of both parameters was consistent with acute inflammation followed with either by recovery or by death. The model predicted that mild thrombocytopenia was present in both survivors and non-survivors.

However, while platelets' level had a steady trend toward normalization in survivors, platelets' counts irreversibly fell during the second week of the diseases in non-survivors, possibly because of a consumption related to the development of micro-thrombotic events in small vessels. The falling platelets' counts in non-survivors was paralleled with a progressive increase of MPV suggesting that the increased platelets' consumption and the production of pro-inflammatory cytokines led to an increased, yet insufficient, megakaryocytic activity with release in the circulation of young, large volume platelets [20]. Our results concur with a recent metanalysis of nine studies with 1779 COVID-19 patients, showing that low platelets' counts were associated with increased risk of severe disease and mortality [21]. The precise mechanisms by which SARS-CoV-2 affects the hematopoietic system in general and platelets in particular are yet unclear. Reduced production, augmented destruction and augmented consumption have been suggested [22,23].

*The third set of CBC parameters* were RBC counts, HB, MCV and RDW. Our model suggested that variation of the four analysed RBC parameters were marginal and with few clinical implications. The level of RBC counts, HB and MCV were generally within normal range and difference found between survivors and non-survivors were most probably due to small systematic differences in the treatment rather than a direct consequence of the COVID-19 itself. Notably model predicted higher RDW levels in non-survivors than in survivors. This observation was consistent with current knowledge that identified RDW as a marker of dysregulated inflammatory response [24], a negative prognostic index for patients with pneumonia [25], and a predictor of severe lung injury [26,27].

The emerged significant associations between potential confounders and CBC parameters might have several different explanations. The association between age and alteration of platelets' and WBCs' parameters (mainly lymphopenia), probably reflects the severity of COVID-19 [5,28]. Instead, the association between age, chronical renal failure and cardiovascular diseases with the alteration of the RBCs' parameters, mainly anaemia and anisocytosis, probably reflects the stage of the chronical conditions [29,30].

Whilst we modelled data by techniques considered robust, previously validated for analysing complex datasets made of sparse repeated measures in different fields of health [31] and clinical infectious diseases research [11,32–34], there are limitations of our study, inherent to observational studies carried out on data collected in real-time inpatients' clinical management. The study population was represented in majority by males (273, 72%) and this might have impacted on the interpretation of the results. In fact, clinical outcomes can be influenced by pre-existing comorbidities, such as hypertension, cardiovascular disease and diabetes, which tended to be more frequent and more severe in men [35]. Another limitation was analysing only CBC parameters without other laboratory/clinical signs of infection, and without considering different treatment provided. Moreover, although we found significant associations between clinical outcome and haematological parameters, a causal relationship cannot be inferred.

Despite the above limitations, our study describes for the first time the association between haematological parameters and COVID-19 clinical outcomes, through the analysis of routinely collected clinical and laboratory data. Although we exercise caution in the interpretation of results using 'routine' laboratory parameters, our findings point out very relevant factors in the dynamic nature of COVID-19 pathophysiology. Poor prognosis is associated with increased neutrophils counts, reduced lymphocytes counts, increased MPV and anaemia with anisocytosis. Even in the absence of deeper analysis of cytokine levels and immune cell subsets, the inverse relationship between neutrophils and lymphocytes shed light on pivotal factors in immune dysregulation and impairment to fight off SARS-CoV-2.These observations have direct implication for future development of more accurate prognostic indexes, composite

outcomes for interventional studies on new drugs and for endorsing new functional studies to assess the role of leukocytes and platelets in the pathogenies of severe infections, such as COVID-19.

## Supporting information

**S1 File. Full STATA statistical analysis.**
(PDF)

**S2 File. Model building and variable selection process.**
(PDF)

**S1 Dataset.**
(XLSX)

## Acknowledgments

The study has been carried out with the support of the INMI COVID-19 Study Group.

Members: Maria Alessandra Abbonizio, Chiara Agrati, Fabrizio Albarello, Gioia Amadei, Alessandra Amendola, Mario Antonini, Raffaella Barbaro, Barbara Bartolini, Martina Benigni, Nazario Bevilacqua, Licia Bordi, Veronica Bordoni, Marta Branca, Paolo Campioni, Maria Rosaria Capobianchi, Cinzia Caporale, Ilaria Caravella, Fabrizio Carletti, Concetta Castilletti, Roberta Chiappini, Carmine Ciaralli, Francesca Colavita, Angela Corpolongo, Massimo Cristofaro, Salvatore Curiale, Alessandra D'Abramo, Cristina Dantimi, Alessia De Angelis, Giada De Angelis, Rachele Di Lorenzo, Federica Di Stefano, Federica Ferraro, Lorena Fiorentini, Andrea Frustaci, Paola Gallì, Gabriele Garotto, Maria Letizia Giancola, Filippo Giansante, Emanuela Giombini, Maria Cristina Greci, Giuseppe Ippolito, Eleonora Lalle, Simone Lanini, Daniele Lapa, Luciana Lepore, Andrea Lucia, Franco Lufrani, Manuela Macchione, Alessandra Marani, Luisa Marchioni, Andrea Mariano, Maria Cristina Marini, Micaela Maritti, Markus Maeurer, Giulia Matusali, Silvia Meschi, Francesco Messina, Chiara Montaldo, Silvia Murachelli, Emanuele Nicastri, Roberto Noto, Claudia Palazzolo, Emanuele Pallini, Virgilio Passeri, Federico Pelliccioni, Antonella Petrecchia, Ada Petrone, Nicola Petrosillo, Elisa Pianura, Maria Pisciotta, Silvia Pittalis, Costanza Proietti, Vincenzo Puro, Gabriele Rinonapoli, Martina Rueca, Alessandra Sacchi, Francesco Sanasi, Carmen Santagata, Silvana Scarcia, Vincenzo Schininà, Paola Scognamiglio, Laura Scorzolini, Giulia Stazi, Francesco Vaia, Francesco Vairo, Maria Beatrice Valli, Alimuddin Zumla.

GI and AZ are co-principal investigators of the Pan-African Network on Emerging and Re-Emerging Infections, which is funded by the EDCTP, the European Union Horizon 2020 Framework Programme for Research and Innovation; and is also in receipt of a NIH Research senior investigator award.

## Author Contributions

**Conceptualization:** Chiara Montaldo, Emanuele Nicastri, Chiara Agrati, Nicola Petrosillo, Andrea Antinori, Vincenzo Puro, Antonino Di Caro, Fabrizio Palmieri, Gianpiero D'Offizi, Luisa Marchioni, Gary Pignac Kobinger, Markus Maeurer, Enrico Girardi, Maria Rosaria Capobianchi, Alimuddin Zumla, Franco Locatelli.

**Data curation:** Simone Lanini, Emanuele Nicastri, Francesco Vairo, Nicola Petrosillo, Paola Scognamiglio, Andrea Antinori, Vincenzo Puro, Antonino Di Caro, Gabriella De Carli, Assunta Navarra, Alessandro Agresta, Claudia Cimaglia, Fabrizio Palmieri, Gianpiero D'Offizi, Luisa Marchioni, Enrico Girardi, Maria Rosaria Capobianchi, Alimuddin Zumla.

**Formal analysis:** Simone Lanini, Chiara Montaldo, Assunta Navarra, Alessandro Agresta, Claudia Cimaglia, Enrico Girardi.

**Funding acquisition:** Giuseppe Ippolito.

**Investigation:** Simone Lanini, Emanuele Nicastri, Francesco Vairo, Chiara Agrati, Paola Scognamiglio, Andrea Antinori, Vincenzo Puro, Gabriella De Carli, Gary Pignac Kobinger, Markus Maeurer, Alimuddin Zumla, Franco Locatelli, Giuseppe Ippolito.

**Methodology:** Simone Lanini, Emanuele Nicastri, Francesco Vairo, Vincenzo Puro, Antonino Di Caro, Gary Pignac Kobinger, Markus Maeurer, Enrico Girardi, Maria Rosaria Capobianchi, Alimuddin Zumla, Franco Locatelli, Giuseppe Ippolito.

**Resources:** Antonino Di Caro.

**Software:** Assunta Navarra, Alessandro Agresta, Claudia Cimaglia.

**Supervision:** Simone Lanini, Giuseppe Ippolito.

**Validation:** Chiara Montaldo, Francesco Vairo, Chiara Agrati, Nicola Petrosillo, Paola Scognamiglio, Andrea Antinori, Vincenzo Puro, Antonino Di Caro, Assunta Navarra, Alessandro Agresta, Claudia Cimaglia, Fabrizio Palmieri, Gianpiero D'Offizi, Luisa Marchioni, Markus Maeurer, Maria Rosaria Capobianchi, Alimuddin Zumla, Franco Locatelli, Giuseppe Ippolito.

**Visualization:** Gabriella De Carli, Franco Locatelli.

**Writing – original draft:** Simone Lanini, Chiara Montaldo, Nicola Petrosillo, Alimuddin Zumla.

**Writing – review & editing:** Simone Lanini, Chiara Montaldo, Emanuele Nicastri, Francesco Vairo, Chiara Agrati, Paola Scognamiglio, Andrea Antinori, Vincenzo Puro, Antonino Di Caro, Gabriella De Carli, Assunta Navarra, Alessandro Agresta, Claudia Cimaglia, Fabrizio Palmieri, Gianpiero D'Offizi, Luisa Marchioni, Gary Pignac Kobinger, Markus Maeurer, Enrico Girardi, Maria Rosaria Capobianchi, Alimuddin Zumla, Franco Locatelli, Giuseppe Ippolito.

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
