## [Decision Letter · Decision Letter 0]

26 Aug 2020

PONE-D-20-21231

COVID-19 Disease - Temporal analyses of complete blood count parameters over course of illness, and relationship to patient demographics and management outcomes in survivors and non-survivors: a longitudinal descriptive cohort study

PLOS ONE

Dear Dr. Lanini,

Thank you for submitting your manuscript to PLOS ONE. After careful consideration, we feel that it has merit but does not fully meet PLOS ONE’s publication criteria as it currently stands. Therefore, we invite you to submit a revised version of the manuscript that addresses the points raised during the review process.

We look forward to receiving your revised manuscript.

Kind regards,

Silvia Ricci

Academic Editor

PLOS ONE

Journal Requirements:

2. PLOS ONE requires experimental methods to be described in enough detail to allow suitably skilled investigators to fully replicate and evaluate your study. See https://journals.plos.org/plosone/s/submission-guidelines#loc-materials-and-methods for more information.

Please describe your RT-PCR assay in more detail.

3. Please replace p-values of "0.000" in your tables to "<0.001".

4. Thank you for stating in the text of your manuscript:

"This study was approved by the IRB of Italian National Institute for Infectious Diseases “Lazzaro Spallanzani” (INMI), in Rome (Italy)".

Please also add this information to your ethics statement in the online submission form.

In addition, please provide additional details regarding participant consent.

In your methods and ethics statement in the online submission form, please ensure that you have specified (i) whether consent was obtained, (ii) whether consent was informed, and (iii) what type you obtained (for instance, written or verbal, and if verbal, how it was documented and witnessed).

If your study included minors, state whether you obtained consent from parents or guardians.

If the need for consent was waived by the ethics committee, please include this information.

6. Please amend your list of authors on the manuscript to ensure that each author is linked to an affiliation. Authors’ affiliations should reflect the institution where the work was done (if authors moved subsequently, you can also list the new affiliation stating “current affiliation:….” as necessary).

7. Please include captions for your Supporting Information files at the end of your manuscript, and update any in-text citations to match accordingly. Please see our Supporting Information guidelines for more information: http://journals.plos.org/plosone/s/supporting-information

Reviewers' comments:

Reviewer's Responses to Questions

**Comments to the Author**

1. Is the manuscript technically sound, and do the data support the conclusions?

Reviewer #1: Yes

Reviewer #2: Yes

2. Has the statistical analysis been performed appropriately and rigorously? 

Reviewer #1: I Don't Know

Reviewer #2: I Don't Know

3. Have the authors made all data underlying the findings in their manuscript fully available?

Reviewer #1: Yes

Reviewer #2: Yes

4. Is the manuscript presented in an intelligible fashion and written in standard English?

Reviewer #1: Yes

Reviewer #2: No

5. Review Comments to the Author

Reviewer #1: Lanini et al present the results of a temporal analyses of complete blood count parameters over course of COVID-19 illness, and relationship to patient demographics and management outcomes in survivors and non-survivors in a large Italian referral center. Please find below my comments:

1) The manuscript should be reviewed by the authors for minor grammar and syntax errors

2) Abstract and Results: please provide also absolute values instead of reporting only p-values

3) Abstract: Please specify the "reverse temporal trends".

4) Abstract: Please define the "average" estimates in the methods section

5) Abstract and Discussion: "Increased neutrophilcounts, reduced lymphocytecounts, median platelet volume, anaemia with anisocytosis, in association with obesity, chronic renal failure,..." Please restate the phrase "in association with"

6) Please provide figures of better quality

7) Please define "history of neoplasm". Does it mean current chemotherapy or those receiving chemotherapy and those under surveillance are included? Do you have any data regarding the patients currently receiving chemotherapy (eg last 14 days from symptom onset) compared to the others?

8) Please expand the Table legends and state that the reported differences are between survivors and non-survivors

9) Please add units in the Tables, as appropriate

10) Please consider supporting the rationale of the study in the introduction and discussion (eg consider this review E Terpos et al. Hematological findings and complications of COVID-19. Am J Hematol . 2020 Jul;95(7):834-847. doi: 10.1002/ajh.25829. Epub 2020 May 23. )

11) Do you have any data regarding CRP and PCT values in different time points in order to correlate them with the corresponding neutrophil counts?

12) In the limitations please make a comment regarding the 72% male percentage in the study cohort and if this has any impact on the interprentation of the results

13) Several significant associations have emerged between potential confounders and CBC parameters, as described in the results. Please comment and elaborate more on potential explanations in the discussion.

Reviewer #2: General comments:

- the manuscript can be shortened substantially, especially the results and the discussion. In fact, the results are presented in a very descriptive way, and are already partly discussed.

- as stated in the background section, “complete blood count provides vital parameters which can inform presence of infection, response to treatment, …”. In the following analysis, laboratory/clinical signs of infection and different treatment were not considered. I suggest highlighting it among the limits of the study.

- editing for grammar and language will be helpful.

Specific comments:

- study design: I am still a bit confused by the protocol - were laboratory tests performed with fixed timing (e.g. once per week)? If not, how many blood tests per patient were considered at least and at most? In fact, there is a very interesting day by day analysis (resumed on the Tables), but there is no mention of the number of tests considered for each single day. I am concerned that this might impact the strength of the results for the first days after symptom onset, when probably just a minority of patients was attended.

- discussion section: “data were collected daily in inpatients with COVID-19 disease until time to discharge or death” – this information is in contrast with what stated in the methods and results section (mean of 4.4 laboratory tests per patient – tested for 21 days after symptom onset).

- conclusions might be presented in a more appropriate fashion: e.g. the first part of the last paragraph (“our study describes for the first time the evolution of COVID-19 illness over time”) is not supported by the data presented. A clear conclusion section might be helpful, with a synthetic resume of the most important findings.

- legends to figures are too long and repetitive.

- tables: there are no explanations for the acronyms used, i.e. llb and ulb. Moreover, I couldn’t find a reference for Table 2 inside the text.

6. PLOS authors have the option to publish the peer review history of their article (what does this mean?). If published, this will include your full peer review and any attached files.

Reviewer #1: No

Reviewer #2: No

---

## [Author Response · Author response to Decision Letter 0]

30 Oct 2020

Reviewer #1: 

Lanini et al present the results of a temporal analyses of complete blood count parameters over course of COVID-19 illness, and relationship to patient demographics and management outcomes in survivors and non-survivors in a large Italian referral center. Please find below my comments:

1) The manuscript should be reviewed by the authors for minor grammar and syntax errors. 

We have reviewed and corrected grammar and syntax errors (corrections marked in track changes in the manuscript). 

2) Abstract and Results: please provide also absolute values instead of reporting only p-values.

According to the request of the reviewer, we have added OR and 95%CI for the bivariable and multivariable analysis results (as for table 1) (current lines: 56 – 62). However, we have left just the p-value of the differences between survivors and non-survivors for temporal analysis (mixed effect models). In fact, these p-values assess whether there is a significant difference in the temporal trend for each individual parameter between survivors and non-survivors as a whole. 

3) Abstract: Please specify the "reverse temporal trends". 

We agree with the reviewer that the statement about “reverse temporal trends” in the abstract is redundant and might be confusing. In fact, the observed different trend has been already reported in the previous sentence. We removed the statement (current line 52).

4) Abstract: Please define the "average" estimates in the methods section. 

In the methods section of the abstract, as per reviewer recommendation, we have defined the model-based punctual estimates, as average of all patients’ values (current lines 43 - 44).

5) Abstract and Discussion: "Increased neutrophil counts, reduced lymphocyte counts, median platelet volume, anaemia with anisocytosis, in association with obesity, chronic renal failure,..." Please restate the phrase "in association with". 

As per reviewer’s suggestion, we have restated the phrase reformulating it as follows: “Increased neutrophil counts, reduced lymphocyte counts, increased median platelet volume and anaemia with anisocytosis, are poor prognostic indicators for COVID19, after adjusting for the confounding effect of obesity, chronic renal failure, COPD, cardiovascular diseases and age >60 years” (current lines 65- 68). 

6) Please provide figures of better quality: 

We have provided all the figures (1, 2, 3, 4) in TIFF format.

7) Please define "history of neoplasm". Does it mean current chemotherapy or those receiving chemotherapy and those under surveillance are included? Do you have any data regarding the patients currently receiving chemotherapy (eg last 14 days from symptom onset) compared to the others? 

As per reviewer recommendation, we have added in the manuscript the definition of “history of neoplasm” defined as “any anamnestic data of oncological disease” (current line 183).

Nineteen patients of the cohort had a positive anamnesis for oncological diseases. None of them received chemotherapy during the admission for COVID-19 and/or in the 14 days before the symptoms’ onset. Out of these 19 patients, 5 died. 

8) Please expand the Table legends and state that the reported differences are between survivors and non-survivors. 

According to the reviewer’s indications, we have expanded the tables’ legends stating that the reported differences are between survivors and non-survivors, in tables 2, 3 and 4.

9) Please add units in the Tables, as appropriate. 

According to the reviewer’s indications, we have added the units in the tables 1, 2, 3 and 4. 

10) Please consider supporting the rationale of the study in the introduction and discussion (eg consider this review E Terpos et al. Hematological findings and complications of COVID-19. Am J Hematol . 2020 Jul;95(7):834-847. doi: 10.1002/ajh.25829. Epub 2020 May 23). 

We have added in the background a sentence supporting the rational of the study, based on the review indicated by the reviewer stating that: “As underlined by a recent review, COVID‐19 has a significant impact on the hematopoietic system: lymphopenia, neutrophil/lymphocyte ratio and peak platelet/lymphocyte ratio may be considered as cardinal laboratory findings, with prognostic potential” (current lines 77 – 80), and we have added the related reference (ref. 5).

11) Do you have any data regarding CRP and PCT values in different time points in order to correlate them with the corresponding neutrophil counts? 

Data regarding CRP and PCT values in different time points would have definitively been extremely interesting. Unfortunately, we didn’t have the possibility to analyze these data, and we have mentioned it as a limitation of the study (current lines 337—339). 

12) In the limitations please make a comment regarding the 72% male percentage in the study cohort and if this has any impact on the interpretation of the results. 

As suggested by the reviewer, in the discussion we have added the higher percentage of males as possible limitation of the study, explaining that “clinical outcomes can be influenced by pre-existing comorbidities, such as hypertension, cardiovascular disease and diabetes, which tended to be more frequent and more severe in men” (current lines 335 – 337) and we have added a related reference (ref 34): Ambrosino I, Barbagelata E, Ortona E, et al. Gender differences in patients with COVID-19: a narrative review. Monaldi Arch Chest Dis. 2020;90(2):10.4081/monaldi.2020.1389. Published 2020 May 25. 

13) Several significant associations have emerged between potential confounders and CBC parameters, as described in the results. Please comment and elaborate more on potential explanations in the discussion.

We thank the reviewer for this remark which allowed us to add potential explanations regarding the association between potential confounders and CBC parameters at the end of the discussion (current lines 324 – 329): “The emerged significant associations between potential confounders and CBC parameters might have several different explanations. The association between age and alteration of WBCs’ and platelets’ parameters, mainly lymphopenia, probably reflects the severity of COVID-19. Instead, the association between age, chronical renal failure and cardiovascular diseases with the alteration of the red blood cells’ parameters, mainly anemia and anisocytosis, probably reflects the stage of the chronical conditions”.

In support of these explanations we have added 3 new references:

27. Linton, P., Dorshkind, K. Age-related changes in lymphocyte development and function. Nat Immunol 5, 133–139 (2004). https://doi.org/10.1038/ni1033

28. M. Ajaimy and M.L. Melamed. COVID-19 in Patients with Kidney Disease. CJASN 15: 1087–1089, Vol 15 August, 2020

29. Clerkin KJ, Fried JA, Raikhelkar J, et al. COVID-19 and Cardiovascular Disease. Circulation. 2020;141(20):1648-1655. doi:10.1161/CIRCULATIONAHA.120.046941

Reviewer #2: 

General comments

1) The manuscript can be shortened substantially, especially the results and the discussion. In fact, the results are presented in a very descriptive way, and are already partly discussed. 

According to the reviewer’s suggestion, we have shortened results and discussion, eliminating the overlapping parts.

2) As stated in the background section, “complete blood count provides vital parameters which can inform presence of infection, response to treatment, …”. In the following analysis, laboratory/clinical signs of infection and different treatment were not considered. I suggest highlighting it among the limits of the study. 

As suggested by the reviewer, we have highlighted that analyzing only CBC parameters without other laboratory/clinical signs of infection, and without considering different treatment provided, represents a limitation of the study (lines 337 – 339).

3) Editing for grammar and language will be helpful. 

We have reviewed and corrected grammar and syntax errors (corrections marked in track changes in the manuscript). 

Specific comments:

4) Study design: I am still a bit confused by the protocol - were laboratory tests performed with fixed timing (e.g. once per week)? If not, how many blood tests per patient were considered at least and at most? In fact, there is a very interesting day by day analysis (resumed on the Tables), but there is no mention of the number of tests considered for each single day. I am concerned that this might impact the strength of the results for the first days after symptom onset, when probably just a minority of patients was attended.

Thank you for this question that permitted us to better emphasize technical issues of the statistical model. As correctly suggested by the reviewer, the number of measures vary among patients. To deal with this issue we have used a mixed effect multilevel model (MEML) for unbalanced samples. This kind of models are among the most robust statistical techniques for dealing with sparse repeated measures that usually come from clinical dataset made of routine collected data. 

In particular our model has a random coefficient for intercept for dealing with correlation at the level of the patients; a random slope coefficient at the level of time for dealing with correlation between time and the random intercept; an unstructured covariance matrix coefficient that is specifically set for dealing with potential additional issues due to unbalanced sampling, consequent to unequal number of measures for the different subjects (i.e no assumption on variance-covariance structure was made). 

We have already validated this model in previous studies that dealt with repeated spare measures:

• Lanini S, Bartolini B, Taibi C, et al. Early improvement of glycaemic control after virus clearance in patients with chronic hepatitis C and severe liver fibrosis: a cohort study. New Microbiol. 2019;42(3):139-144.

• Lanini S, Portella G, Vairo F, et al. Relationship Between Viremia and Specific Organ Damage in Ebola Patients: A Cohort Study. Clin Infect Dis. 2018;66(1):36-44. doi:10.1093/cid/cix704

• Lanini S, Portella G, Vairo F, et al. Blood kinetics of Ebola virus in survivors and nonsurvivors. J Clin Invest. 2015;125(12):4692-4698. doi:10.1172/JCI83111

According to the reviewer’s suggestion, statistical methods has been expanded to report number of observations per patients and per time point (current lines 141 – 143). 

5) Discussion section: “data were collected daily in inpatients with COVID-19 disease until time to discharge or death” – this information is in contrast with what stated in the methods and results section (mean of 4.4 laboratory tests per patient – tested for 21 days after symptom onset).

We thank the reviewer to have highlighted an incorrect sentence that gave an inconsistent message. In fact we have analyzed the tests performed within the first 21 days after the symptoms’ onset (not until time of discharge or death). According to the reviewer’s comment, we have removed the first sentence since it is incorrect and since the correct information is already provided in the method section (current lines 140 – 143).

6) Conclusions might be presented in a more appropriate fashion: e.g. the first part of the last paragraph (“our study describes for the first time the evolution of COVID-19 illness over time”) is not supported by the data presented. A clear conclusion section might be helpful, with a synthetic resume of the most important findings. 

We have edited the conclusion according with the reviewer suggestions as follows (lines 342 – 353): “Despite the above limitations, our study describes for the first time the association between haematological parameters and COVID-19 clinical outcomes, through the analysis of routinely collected clinical and laboratory data. Although we exercise caution in the interpretation of results using ‘routine’ laboratory parameters, our findings point out very relevant factors in the dynamic nature of COVID-19 pathophysiology. Poor prognosis is associated with increased neutrophils counts, reduced lymphocytes counts, increased MPV and anaemia with anisocytosis. Even in the absence of deeper analysis of cytokine levels and immune cell subsets, the inverse relationship between neutrophils and lymphocytes shed light on pivotal factors in immune dysregulation and impairment to fight off SARS-CoV-2. These observations have direct implication for future development of more accurate prognostic indexes, composite outcomes for interventional studies on new drugs and for endorsing new functional studies to assess the role of leukocytes and platelets in the pathogenies of severe infections, such as COVID-19”.

7) Legends to figures are too long and repetitive. 

As suggested by the reviewer, we have shorten the legends of the figures deleting repetitive parts already mentioned in the main article.

8) Tables: there are no explanations for the acronyms used, i.e. llb and ulb. Moreover, I couldn’t find a reference for Table 2 inside the text. 

As suggested by the reviewer, we have added a legend to explain the acronyms in tables 2, 3 and 4: llb= low limit bound, ulb= lower limit bound.

The reference to table 2 is in the results section in the chapter: Leukocytes parameters kinetics (current line 192).

---

## [Decision Letter · Decision Letter 1]

4 Dec 2020

COVID-19 Disease - Temporal analyses of complete blood count parameters over course of illness, and relationship to patient demographics and management outcomes in survivors and non-survivors: a longitudinal descriptive cohort study

PONE-D-20-21231R1

Dear Dr. Simone Lanini

We’re pleased to inform you that your manuscript has been judged scientifically suitable for publication and will be formally accepted for publication once it meets all outstanding technical requirements.

Kind regards,

Silvia Ricci

Academic Editor

PLOS ONE

Additional Editor Comments (optional):

Reviewers' comments:

Reviewer's Responses to Questions

**Comments to the Author**

1. If the authors have adequately addressed your comments raised in a previous round of review and you feel that this manuscript is now acceptable for publication, you may indicate that here to bypass the “Comments to the Author” section, enter your conflict of interest statement in the “Confidential to Editor” section, and submit your "Accept" recommendation.

Reviewer #1: All comments have been addressed

Reviewer #2: All comments have been addressed

2. Is the manuscript technically sound, and do the data support the conclusions?

Reviewer #1: (No Response)

Reviewer #2: Yes

3. Has the statistical analysis been performed appropriately and rigorously? 

Reviewer #1: (No Response)

Reviewer #2: I Don't Know

4. Have the authors made all data underlying the findings in their manuscript fully available?

Reviewer #1: (No Response)

Reviewer #2: Yes

5. Is the manuscript presented in an intelligible fashion and written in standard English?

Reviewer #1: (No Response)

Reviewer #2: Yes

6. Review Comments to the Author

Reviewer #1: (No Response)

Reviewer #2: Thank you for addressing all the comments.

Still, I do not feel familiar with the acronyms you used in the Tables: llb= low limit bound, ulb= lower limit bound. Standing on the numbers reported, it looks like if "ulb" represented the upper limit bound. In any case, I would suggest to further explain the significance of llb and ulb in that context.

7. PLOS authors have the option to publish the peer review history of their article (what does this mean?). If published, this will include your full peer review and any attached files.

Reviewer #1: No

Reviewer #2: No

---

## [Editor Report · Acceptance letter]

16 Dec 2020

PONE-D-20-21231R1 

COVID-19 Disease - Temporal analyses of complete blood count parameters over course of illness, and relationship to patient demographics and management outcomes in survivors and non-survivors: a longitudinal descriptive cohort study 

Dear Dr. Lanini:

I'm pleased to inform you that your manuscript has been deemed suitable for publication in PLOS ONE. Congratulations! Your manuscript is now with our production department. 

Kind regards, 

on behalf of

Dr. Silvia Ricci 

Academic Editor

PLOS ONE